# Is massage a legitimate part of nursing care? A qualitative study

**Gora Da Rocha Rodrigues**[1,2☯]*, **Adrien Anex**[2☯], **Monique Boegli**[3‡], **Catherine Bollondi Pauly**[4‡], **François Curtin**[5‡], **Christophe Luthy**[6‡], **Jules Desmeules**[3‡], **Christine Cedraschi**[3,6,7☯]

**1** HEdS-School of Health Sciences, HES-SO University of Applied Sciences and Arts Western Switzerland, Geneva, Switzerland, **2** HESAV-School of Health Sciences, HES-SO University of Applied Sciences and Arts Western Switzerland, Lausanne, Switzerland, **3** Division of Clinical Pharmacology and Toxicology, Multidisciplinary Pain Centre, University Hospitals of Geneva, Geneva, Switzerland, **4** Professional Practices Pole, Care Direction, University Hospitals of Geneva, Geneva, Switzerland, **5** University Hospitals, Geneva, Switzerland, **6** Division of General Medical Rehabilitation, University Hospitals of Geneva, Geneva, Switzerland, **7** Geneva Faculty of Medicine, Geneva, Switzerland

☯ These authors contributed equally to this work.
‡ MB, CBP, FC, CL and JD also contributed equally to this work.
* gora.darocha@hesav.ch

**Data Availability Statement:** Regarding data availability, the data for this study consists of transcripts of two focus groups that contain identifying information. The data cannot be shared

## Abstract

### Introduction

The use of massage therapy has received increased attention in the treatment of chronic pain. However, barriers can hinder its use in nursing care. This study uses a qualitative methodology to explore professionals' experiences regarding touch massage (TM) and identify barriers and facilitators for the implementation of this intervention.

### Materials and methods

This study is part of a larger research program aimed at investigating the impact of TM on the experiences of patients with chronic pain hospitalized in two units of an internal medicine rehabilitation ward. Health care professionals (HCPs) were trained either to provide TM or to use of a massage-machine device according to their units. At the end of the trial, two focus groups were conducted with HCPs from each unit who took part in the training and agreed to discuss their experience: 10 caregivers from the TM group and 6 from the machine group. The focus group discussions were tape-recorded, transcribed and analyzed using thematic content analysis.

### Results

Five themes emerged from thematic content analysis: perceived impact on patients, HCPs' affective and cognitive experiences, patient-professionals relationships, organizational tensions, and conceptual tensions. Overall, the HCPs reported better general outcomes with TM than with the machine. They described positive effects on patients, HCPs, and their relationships. Regarding interventions' implementation, the HCPs reported organizational barriers such as patients' case complexity, work overload, and lack of time. Conceptual barriers

publicly due concerns of healthcare professional confidentiality and ethics requirements. Interviews were confidential to enable freedom of expression by participants, and participants consented to the study with the understanding that only de-identified quotations would be made public, not the entirety of the transcripts. Therefore, only illustrative quotes from the transcripts have been included in this paper. Data access requests may be made to the Regional Ethics committee at ccer@etat.ge.ch.

**Funding:** GDR-JD-CBP-CC: Leenaards Fondation (Grant number 4518.1) https://www.leenaards.ch/ CBP-MB: Fonds de développement de la Direction des soins HUG" https://www.hug.ch/direction-medicale-qualite/recherche-clinique The funders had no role in study design, data collection and analysis, decision to publish, or preparation of the manuscript.

**Competing interests:** The authors have declared that no competing interests exist.

such as ambivalence around the legitimacy of TM in nursing care were reported. TM was often described as a pleasure care that was considered a complementary approach and was overlooked despite its perceived benefits.

## Conclusion

Despite the perceived benefits of TM reported by the HCPs, ambivalence arose around the legitimacy of this intervention. This result emphasizes the importance of changing HCPs' attitudes regarding a given intervention to facilitate its implementation.

## 1. Introduction

Chronic pain is a major health issue across the world and affects around 20% of the general population [1, 2]. The International Classification of Diseases 11th Revision (ICD-11) defines chronic pain is defined as "persistent or recurrent pain lasting longer than 3 months" [3]. The high prevalence of chronic pain calls for clinicians and finding appropriate treatment to improve life of patients. To this end, the use of massage therapy has increased in chronic pain treatment [1, 4]. Massage therapy has been reported to be effective for reducing delayed onset muscle soreness and postoperative, labor, low back, cancer-related, and musculoskeletal pain, among other types [5]. Furthermore, massage therapy can reduce discomfort from various conditions, such as fibromyalgia [6], chronic low back pain [7, 8], or chronic pain [9].

Aims, intentions, and techniques differ greatly in massage therapy [10], making comparison challenging. However, its effects on pain reduction are low to moderate. In addition to pain reduction, massage therapy has been associated with depressive symptom reduction [6, 11, 12], anxiety reduction [6, 11–14], increased well-being [11, 12], and treatment satisfaction [13, 15]. Overall, the benefits of massage therapy make it a promising complementary or alternative medicine (CAM) treatment for patients with chronic pain. Nonetheless, further research is needed on how to implement such interventions in nursing care.

The favorable attitudes of health care professionals (HCPs) toward CAM are well documented [16]. HCPs have reported positive effects of CAM on their job satisfaction, care provided, and patient-provider relationships [17]. However, its use is limited [17–19]. Barriers such as the lack of knowledge, trained staff, evidence, and time can hinder its integration in general care [16, 17, 20]. Massage therapy is one of the types of CAM most used by HCPs [16, 18, 20]. Similar to CAM, HCPs have reported favorable attitudes toward the use of massage in giving care [21, 22] and have described positive effects such as enhanced patient-provider relationships [23, 24], increased sensitivity and attentiveness to the patient [25], and relaxation [26]. Overall, CAM and more specifically massage are well accepted by health professionals and are often associated with more holistic approaches to care [21].

Trends focused on illnesses and cures have long prevailed in nursing education and research. Biomedical protocols and quantitative research have become dominant to the detriment of health promotion and well-being-oriented nursing interventions. Since the 1970s, the nursing discipline has proposed a new paradigm with an approach that considers wholeness [27]. This human-centered care has a unitary vision and integrates the concept of well-being [28]. The unitary-transformative paradigm brings a new perspective, but for many caregivers, their delegated roles take precedence over their autonomous roles.

The present study focuses on HCPs' experiences regarding a specific massage intervention called touch massage (TM) [29]. In recent years, specialized nurses at a large Swiss tertiary

university hospital have received training on this method. TM has been defined as "a benevolent intention that takes shape through touch and the sequence of movements on all or parts of the body, that allows relaxation, fitness, reassurance, communication or simply well-being, pleasant to receive and, what is more, to practice" [29].

Previous research on TM and its effects on chronic pain conducted at the same hospital highlighted reductions in pain intensity along with other symptoms, improvements in patient well-being, and better patient-professional relationships [30–32]. Drawing from the literature and from previous experience, we assumed that TM can have positive impacts on patients with chronic pain. Nevertheless, little is known regarding the experiences of HCPs who used this method. This study is part of a larger research program aimed at investigating the impacts of TM on the experiences of patients with chronic pain hospitalized in an internal medicine rehabilitation ward. In a first step, we conducted a non-randomized cluster trial with the patients. The present study is a second step using a qualitative methodology to explore professionals' experiences regarding TM and identify limits to and facilitators for the implementation of this intervention.

## 2. Materials and methods

### 2.1 Setting

The study was conducted at a 90-bed university-based general medical rehabilitation ward comprising five units of 18 patients each. The units are similar in terms of care and population. The ward is part of a 1,200-bed urban public and teaching hospital that is the major primary care hospital for the area and is devoted to general medical rehabilitation and psychosocial care with a specific emphasis on comprehensive active rehabilitation and multidisciplinary treatment. Patients are either transferred from acute care wards (about 2/3) or directly admitted from the emergency room (about 1/3) to any one of the five units in the ward. A vast majority of the patients are discharged home, 7–8% of the patients die during their stays in the ward, and only very few (1–2%) require definitive institutionalization. The median and mean length of stays were 16 and 21 days, respectively, during the study period (between October 2019 and June 2020).

The study was designed as a non-randomized cluster clinical trial with an exploratory qualitative part [32]. Two units of a general rehabilitation ward were selected for this trial. TM has been assigned to one unit (intervention group; IG) whereas machine-delivered massage has been assigned to the other (control group; CG). Patients were allocated to the units following the usual general allocation rules of the ward. The HCPs received training on TM or the use of the massage-delivering machine according to their unit (CG or IG). The interventions were administered by trained nursing auxiliaries and nurses on the care team of each unit.

This study received the approval of the Cantonal Commission for Ethics and Human Research in Geneva (CCER 2019–00848) and was pre-registered (ClinicalTrials.gov, NCT04295603) [32].

### 2.2 Data collection

At the end of the trial, the HCPs from the two units (CG and IG) involved in the study were invited to participate in two focus groups, one for the CG and one for the IG. The HCPs were trained to provide TM (IG unit) or to use the massage machine (CG unit). The focus groups with the HCPs were conducted by two members of the research team (GD and CBP) trained in interview procedures [33, 34].

HCPs were included if they were working in one of the two selected units and participated in the trainings whether or not they delivered the interventions. We aimed to include as many

opinions as possible as our goal was to investigate the feasibility of the interventions. From the 21 HCPs working in the IG unit, 15 HCPs (7 nurses and 8 nursing auxiliaries) took part in the TM training and in the trial. In the CG unit all 21 HCPs working in this unit were offered participation, and 8 of them took part in the machine-delivered massage training and in the trial. One focus group convened 10 caregivers from the IG unit, and the second six caregivers from the CG unit. Both groups were composed of nurses and nursing auxiliaries who participated in the clinical trial. The focus groups were conducted at the participants' workplace [32] A research assistant was present to help with the recording and take notes for the logbook. Both groups of HCPs qualified as informed respondents.

Focus groups have been chosen to explore the satisfaction and general perception of massage in the multidisciplinary health care teams in the two units concerned. The choice of this method was of relevance in the context of the various units of the ward that all function as specific teams. Thus, a method of data collection that simultaneously generates data for three levels of analysis: the individual, the group and the interactions between participants was of clear interest. The protocol insisted on the need to recruit at least five participants, including nurses, assistant nurses, physical therapists and/or physicians. Based on the experience of our research team regarding TM, an interview guide was devised to assess the impact of massage on general care, the experience thereof, the positive and negative effects, and its impact on the development and planning of care (see Table 1). The interview has been audio-recorded, and transcribed verbatim.

The interview guide explored the HCPs' experiences and use of the interventions (see Table 1). Four dimensions were investigated: recall of the massage, general appreciation of massage, facilitators and barriers in the experience of massage, and benefits for other. Informed consents were collected at the beginning of each focus group. The focus groups were audio-recorded and transcribed verbatim, anonymized, and checked for quality by a member of the research team (GD). The focus groups lasted 1h 30 min for each group (IG and CG).

## 2.3 Data analysis

Analysis of the qualitative data was done using MAXQDA 2022 [35]. Thematic analysis was used to analyze the verbatim transcripts of the focus groups, following the steps described by Braun and Clarke [36, 37]: 1) becoming familiar with the data, 2) generating initial codes, 3)

**Table 1. Interview guide for the focus groups.**

| Themes | Probes |
| --- | --- |
| 1. Recall of massage | What was it like for you to conduct the massage? How did you experience these situations? How did you handle positive or negative emotions arising during the massage? Were there any experiences that made it easier to intervene? |
| 2. General appreciation of massage | What overall feeling do you have left? What were the favorable vs. unfavorable elements? What were your recommendations? |
| 3. Facilitators and barriers in the experience of massage | In terms of organization, what were the facilitating elements? What were the barriers? How did you deal with the difficulties? What resources did you mobilize? (Were they personal, organizational, or patient resources?) How would you feel about introducing this intervention into practice? In terms of feasibility and acceptability, do you think that the caregiver's personal beliefs or experiences influence the acceptability of the intervention? |
| 4. Benefits for others | How do you think the patient experienced the massage? Did they tell you anything related to the massage? Is there anything else you would like to add? |

searching for themes, 4) reviewing themes, 5) defining and naming themes, and 6) producing the report. Data were coded and analyzed by three coders separately (AA, GD, CC). Periodic meetings between coders were held to resolve discrepancies and reach consensus at each step of the analysis. The codes and themes were data driven. To create and define themes and sub-themes, the researchers used MindMeister®, an online tool for mind mapping.

## 3. Results

Five themes emerged from the thematic analysis: perceived impacts on patients, HCPs' affective and cognitive experiences, patient-professional relationships, organizational tensions, and conceptual tensions. Those themes were regrouped into two dimensions: outcomes of the interventions and implementation in nursing care (see Fig 1).

### 3.1 Outcomes of the interventions

**3.1.1 Perceived impacts on patients.** The HCPs described various impacts of the interventions reported by patients. Some HCPs reported the TM had positive effects on patients such as feelings of relaxation or increases in confidence. In contrast, the perceived impacts of the machine on patients were more heterogeneous. According to the HCPs, some patients felt discomfort with the use of the machine and others asked to keep using it.

Patients' feelings of relaxation right after receiving TM were described as improving their sleep. Relaxation was also perceived as leading the patient to feel more confidence in the caregiver.

*"The patient was so happy and then he told me, 'You know, sometimes I get up to go have coffee, and that night I stayed in bed and slept.'" (249–250, IG)*

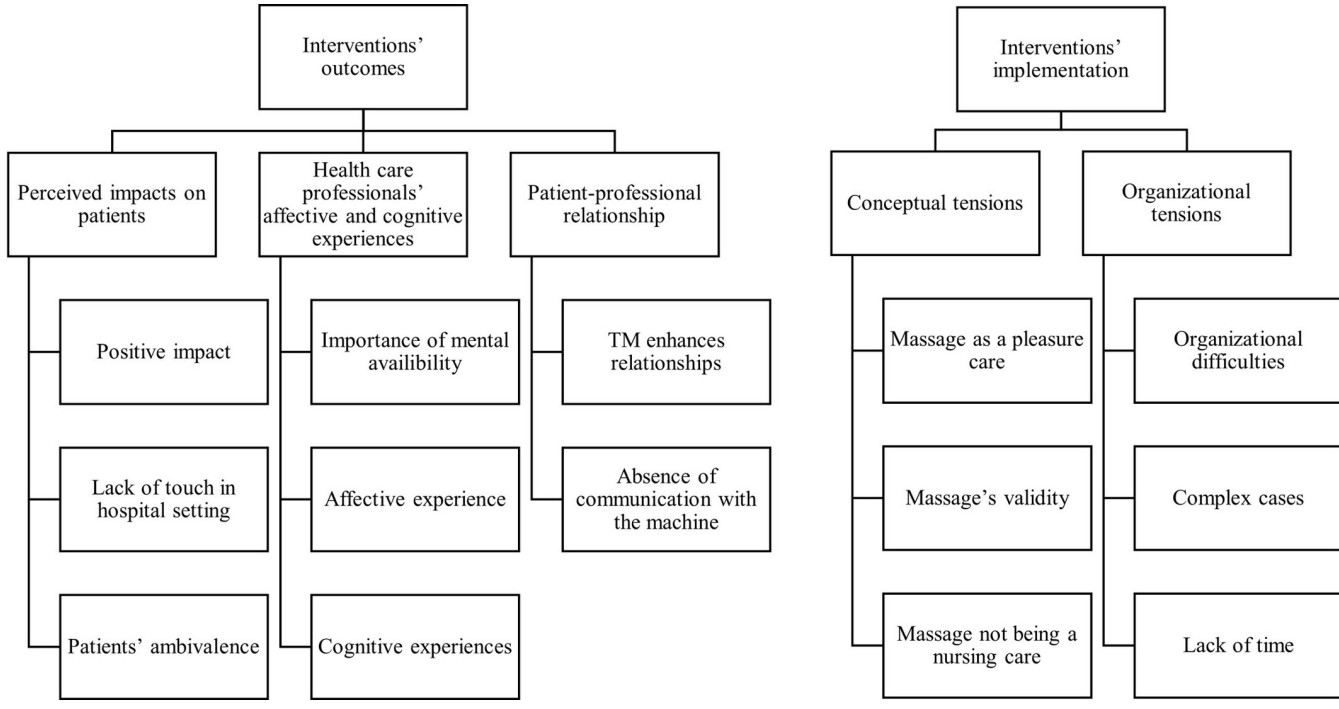

**Fig 1. Thematic categorization.** Dimensions, themes, and sub-themes.

*"I think that it helps the patients a little bit to free themselves to speak; the fact of being a little more relaxed, I think it gives them a little more confidence, and that's important." (242–244, IG)*

The HCPs further raised the question of touch in the hospital context. They deemed it important in a care setting and felt it could have beneficial impacts on patients such as reassuring them. However, the HCPs also reported that sometimes touch is absent in daily care practices.

*"I would say that touch in general brings something to the patient, just touching the hand, the shoulder to reassure. We can see that it benefits them because some have been in hospital for 3–4 months, and nobody has touched them." (227–229, IG)*

*"It's the human relationship: We take the time; we have a physical contact, an exchange. I think they are more valued, and I think that in the end, it is good" (233–234, IG)*

The HCPs reported that the machine was appreciated by some patients but not others. The dislike of the machine reduced patients' motivation to receive the intervention or their desire to use it any further. One of the first reasons mentioned was the fact that patients felt discomfort when using it.

*"I didn't have just one, I had several, and after a while I thought, well, is it worth offering? Because they try once, they feel pain, and they don't want to try it a second time." (68–70, CG)*

However, some patients did not share this discomfort. One nurse stated, *"Well, I remember that a patient told me that she was going to buy it for her personal use at home" (104–105, CG)*.

**3.1.2 Health care professionals' affective and cognitive experiences.** The interventions induced various experiences (emotions and thoughts) in the HCPs and those influenced their motivation to continue giving the interventions to patients. While the HCPs in the IG emphasized a need for mental availability and a surge of emotions, the HCPs in the CG described concerns related to the use of the machine.

In the IG, participants described the importance of the mental availability of the massage provider. The HCPs often emphasized the importance of being in a calm and benevolent state of mind to perform TM.

*"You have to be relaxed to be able to bring relaxation as well." (330, IG)*

*"Mind at rest and the mind totally focused on the patient." (445, IG)*

In contrast, the HCPs described that if they were stressed or tense, the patient would feel it, and it would hinder the process. The HCPs explained this effect in terms of positive or negative energy they could transmit to patients. Furthermore, positive emotions such as feelings of relaxation and having rewarding experiences arose in the HCPs from the TM intervention. The positive impact of TM on the HCPs enhanced their willingness to give TM. However, when giving TM was impossible due to a lack of time or availability, the HCPs felt frustrated.

*"They feel it when we are stressed; they tell us, 'Ah, today you are different.'" (461–462, IG)*

*"I found it frustrating not to do it at all, [. . .] as my colleague said, for me, it's a pleasurable care that gives pleasure to the patient, and it also gives me pleasure to give him this care." (77–79, IG)*

The TM intervention also elicited negative emotions such as disgust, discouragement, or guilt and led the HCPs to seek strategies to reduce those feelings.

*"I had a lady who doesn't wash herself, so it's difficult to do these foot massages, [. . .] so I washed her. It was a lady who didn't want to shower, so I washed her feet anyway before doing the massage because it smelled bad." (32–36, IG)*

*"One time, it discouraged me because the patient, I visited him on the third day, asking him the question, 'Did you like the massage?' and he told me, 'What massage?' He had completely forgotten; I had taken the time to massage him, and on the third day, he had forgotten, so I said it's useless." (275–278, IG)*

In the CG, negative perceptions arose from the use of the machine. The HCPs did not feel comfortable to further press patients to participate in the study and use the machine. In addition, they described the installation of the machine as burdensome and time consuming. The HCPs argued that the machine did not decrease their amounts of work but instead added more tasks for them to do. This caused the HCPs to feel like technicians when using the machine. They compared it with massage, which they considered a familiar procedure, therefore requiring less time. All this strengthened the participants' lack of motivation to use the machine.

*"Well, as I imagine, it's a technician's time, it doesn't make me want to practice this care, it's not a treatment, it's an installation, that's it." (245–247, CG)*

*"I think that yes, compared to the caregivers, it's more, they're better at massaging than going to get a machine, setting it up, doing all that, I think. You must go and get the key, get the machine, put the key back; it's a whole little process. It's a whole process that takes time and you have to think about it." (217–220, CG)*

**3.1.3 Patient-professional relationships.** Changes in the relationship between professional and patient were highlighted through changes in interaction and communication. Communication is described in terms of the exchange of information between two or more individuals. Interaction can be described as a process whereby one's action influences the action of another person (mutual influence). TM was reported as facilitating communication between the patients and HCPs in IG, whereas little to no communication was described in the CG. Furthermore, in the IG, the HCPs noticed changes in their interactions with patients.

The HCPs in the IG reported having discussions during TM. The HCPs described touch as facilitating their interaction with the patient. They also described TM as leading to a different interaction than they experienced in "standard" nursing care. They described those exchanges as being easier thanks to the feelings of relaxation and confidence.

*"It's the human relationship; we take the time, we have a physical contact, an exchange. I think they are more valued, and I think that in the end, it is good because it stimulates other emotions, and it feels good." (233–235, IG)*

*"It's an exchange, and the approach of the care is different than the toilet. It has nothing to do, it is easier with them, they feel more comfortable, more relaxed." (19–20, IG)*

However, in the CG, most of the HCPs reported a lack of communication with their patients. They explained that installing the machine did not lead to any interactions with the patients, making communication difficult. The HCPs opposed the machine compared to the

massage they were doing before bedtime, during which they could better communicate about the patients' care.

*"In the end, we don't share much with the patient; it doesn't necessarily make you want to do it." (227–228, CG)*

*"In the evening, we still had to remove the stockings, rub and massage as well, so you get direct feedback from the patient, who says, 'Well, I prefer this kind of massage,' whereas [with] the machine, they don't say anything; they just say it hurts, is it possible to remove it? But there is no more communication." (146–150, CG)*

### 3.2 Implementation in nursing care

**3.2.1 Conceptual tensions.** One barrier to the implementation of TM or the massage machine in nursing care was the attitudes of the HCPs toward them. In the CG, the machine was described as nothing more than a machine, and ambivalence around the legitimacy of TM in nursing care was reported. In the IG, the HCPs described TM as a pleasure care, which is opposed to "standard" nursing care. The legitimacy of TM in nursing care was questioned, and the perception of TM as a means of care was at best ambivalent. This, in turn, led the HCPs to perceive TM as not a priority and often to set aside the intervention. However, if conditions such as validation of the hierarchy and scientific or empirical evidence were met, TM could be considered a potential nursing care. In the CG, the machine was perceived as bringing relaxation or discomfort but was not mentioned as a care.

Most of the HCPs in the IG described TM as a pleasure care because of the perceived pleasure felt by the patients and care providers during the intervention. The HCPs contrasted this care with other "standard" nursing care and described TM as differing in the way patients and HCPs feel.

*"So, in fact, it's not a medical care; it's a pleasure care, I would say. We see the person differently, and the person has the impression of forgetting their illness." (17–18, IG)*

*"We can also convey something and not just be [. . .] purely medical, convey something else." (377–378, IG)*

*"[TM is] other care that is not just care, such as taking blood pressure, but a benevolent care." (572–573, IG)*

TM was also described as a complementary approach that could be implemented in other care. The HCPs reported using the gestures of TM during the time before bed. Not only did the HCPs use TM with patients, but they also reported using it among themselves or in their private lives.

*"At night, when we do our nursing, we'll have the gestures to put the cream on the legs and arms properly, too. I would say that it has brought something." (174–176, IG)*

*"I would like to say that yes, in addition, it is also good between us because we massage each other. In fact, at the computer, it's also beneficial; it brings us closer together." (216–217, IG)*

*"I found it good personally because I learned something new that I use with my son." (644–645, IG)*

TM was perceived by the HCPs as not being a priority compared to other nursing care. This was further illustrated by the concerns the HCPs expressed over not being available for primary care during TM sessions. This caused some HCPs to overlook the intervention despite having taken part in the training.

> *"I'm not going to leave my colleagues alone with an overloaded service, so yeah, the priority at one point was not massages, frankly." (147–148, IG)*

> *"So, I'm a bit divided because at the same time, I tell myself that it must not be easy to find the right moment if the colleagues need help and we are doing the massage;, it's true that we can't stop it just like that" (79–82, IG)*

> *"So, I think that in fact it's not a question of unwillingness, because well, doing the training was on a voluntary basis, so well, um, we all came, we all wanted to do it, to do it well, and we didn't have the opportunity to practice it." (142–145, IG)*

However, when TM was acknowledged and validated by superiors and empiric evidence, it is described as an acceptable care. This change in attitude lead to feel more comfortable to devote time and attention to TM.

> *"And also, that it's validated by the superiors [. . .]. Validated [by the supervisor], yeah, we are doing right." (397, IG)*

> *"The fact that there is a study on it also I think shows that there is research behind it and that there are effects behind it that are expected." (288–289, IG)*

> *"Since it is accepted as care, we don't have to feel guilty about saying, well, I'm doing a massage while my colleagues are doing the dishes or whatever." (532–534, IG)*

**3.2.2 Organizational tensions.**   This theme grouped the descriptions of organizational barriers encountered in implementation of the interventions. The HCPs in both the CG and IG reported barriers related to patients, colleagues, the organization of the institution, and time.

The HCPs reported that the large number of patients per health care staff member made the intervention difficult without neglecting other patients. The HCPs further described the complexity of their patients' cases and pathologies. They contrasted it with the context of a rehabilitation unit, which in theory, should have fewer complex cases.

> *"We have patients who are chronic, patients who are complex cases. Yeah, it's difficult." (180–181, IG)*

> *"As we organize our work differently, we should have lighter patients who really require re-education and rehabilitation, patients that we had 20 years ago. Now we have acute care, people who take a lot of our time; their health condition is very precarious, so that's also time consuming." (565–569, IG).*

In addition, the HCPs described the limited number of health care staff to take care of the patients. This led to work overloads, which, in turn, led to stress and exhaustion in the HCPs. Problems related to the institutional organization were also mentioned, such as changes in organization, a lack of information regarding the intervention, or the roles of the staff.

*"We have a new organization too; it's . . . we have to adapt, but it's also difficult. It's not a change of organization that makes us less tired or that we have more time for our patients." (182–185, IG)*

*"I don't think we've all been involved in this. I, for example, never had information about this machine. I think a lot of us didn't necessarily." (183–184, CG)*

Finally, time was a concern regarding these interventions. The HCPs in both the CG and IG described that because of time constraints (which were rendered even more salient by the COVID-19 pandemic), they could not do the intervention.

*"There are days when we have a little bit of time, so we'll say to ourselves, 'Let's take advantage of this,' and there are days when we don't even have time to sit down to eat." (86–87, CG)*

*"Frankly, I didn't have the time. There was a period that was a bit overloaded, and we finished the evening rounds; we just had time to finish, to make the transmissions, and the colleague would arrive." (130–132, IG)*

They further reported that the time allocated to the intervention was too long.

*"It's a little process that seems to take time, and you must think about it. You take at least 15 minutes each time to prepare it, I think, and then you must install it, come back 15 minutes later; you must take it back." (219–222, CG)*

*"I didn't do the 15 minutes; that's not possible. It's extremely complicated to find that time, to be available at all levels, to be in front of the patient and find 15 minutes to massage." (169–172, IG)*

However, the care providers in the IG came up with a suggestion to improve the implementation of TM. The HCPs said it would be easier to implement TM at certain times of the day, such as in the afternoon or evening before the patients go to sleep. Indeed, the HCPs reported the importance of having a calm setting to perform TM. Therefore, this time of the day would allow them to have an appropriate setting.

*"For me, it would be the afternoon, not the morning shift, because the morning is often busy, running around." (314–315, IG)*

*"In the afternoon, or maybe it would be a little complicated, but at bedtime for a little bit, to do good before the night, it could also be a good idea." (321–323, IG)*

As for the machine, the suggestion was to have it *"self-service, more of a self-service style, like the bikes they have in the unit, for example; they can go to their sessions by themselves" (92–93, CG).*

## 4. Discussion

The aim of this study was to better understand HCPs experiences with TM. The thematic analysis highlighted five themes that could be split into two categories: interventions' outcomes and implementation in nursing care. Three main outcomes were reported by the HCPs. First, the HCPs said that TM increased the comfort and confidence of patients, whereas patients' attitudes toward the machine were more negative. Second, the HCPs using TM emphasized the importance of mental availability when giving TM and the positive or negative emotions

elicited by it. In the CG, the HCPs described concerns related to the use of the machine, such as discomfort when patients were unwilling to continue, difficulties with the installation, and feeling like a technician. Third, the HCPs described improvements in their interactions and communication with patients when using TM, whereas little to no communication or interactions were reported when using the machine.

Overall, the HCPs seemed to report better outcomes with TM than with the machine. They described positive impacts on patients, the HCPs, and their relationships. This is in line with the literature, where other studies on massage have highlighted the positive impacts of such an intervention on patients' feelings of relaxation [25] or on the patient-professional relationship [21, 24, 26].

Regarding the implementation of the interventions, barriers and facilitators were reported. The HCPs faced two type of barriers that hindered their use of the interventions. The first type of barrier was organizational: In both groups, the HCPs described problems regarding the complexity of their patients' cases, work overloads, the organization, and a lack of time to do the intervention. Similar to these findings, lack of time [17] and organizational difficulties [16] are often reported to hinder the use of CAM.

Second, ambivalence around the legitimacy of TM in nursing care was reported. The HCPs described it as pleasure care, which contrasted with "standard" nursing care. Consequently, the HCPs considered TM a complementary approach that was overlooked despite its perceived benefits. Regarding the machine, the HCPs described it as nothing more than a machine. The positive attitudes of HCPs toward massage [22, 26] or more generally toward CAM [16] are often described in the literature. However, questions regarding the place of massage in nursing care remain. One study explored HCPs' experiences with massage in a pediatric setting [21]. In line with our results, they highlighted the HCPs' ambivalence toward the massage intervention as contrasting the biomedical and holistic perspective. From the biomedical perspective, massage was not seen as a nursing task, although such an intervention could find its place in a holistic perspective. The authors further reported that such ambivalence could be highlighted by the little importance granted to massage compared to other nursing tasks. In this study, the HCPs often reported that TM was not a priority or that they did not have time for the intervention, which could suggest little importance being given to the intervention compared to other nursing care tasks. In the latter case, the questions of time, priority, or not being available for nursing tasks such as taking blood pressures or giving medicines were less likely to be reported.

However, our results further highlighted the importance of the empirical and hierarchical validation of TM, which could facilitate the implementation of this intervention. Indeed, receiving hierarchical approval and empirical evidence made the HCPs more likely to accept TM. The lack of knowledge, hierarchical validation, and empirical evidence were often mentioned as barriers to the implementation of CAM in nursing care [16]. Therefore, adequate trainings for HCPs and supervisors could facilitate the use of TM in nursing care. The HCPs further gave recommendations on how to implement TM during the day. This emphasized the importance of including health care staff in the development of an intervention. Indeed, encouraging autonomy and feelings of competence by including the health care team in the process of developing an intervention can further increase their intrinsic work motivation, which, in turn, can facilitate implementation of the intervention [38]. In this context, studies such as this one are important to contribute more toward making massage an integral part of nursing care.

Despite a paradigm shift in the nursing discipline, it would seem that for some health professionals, quantitative evidence and evidence-based medicine are still present enough in nursing minds to legitimize their care. The more "autonomous and holistic" care that integrates

touch as care seems not yet to be shared in a common way. We therefore recommend that nurses continue to be made aware, from their initial training onward, of the added value felt not only by the patient but also by the professionals who deliver this care as legitimate care. Evidence-based nursing should be reinforced through qualitative research.

This study has limits that need to be acknowledged. First, the recruitment period took place during the COVID-19 pandemic. The pandemic added significant duties to HCPs' workloads. This increased HCPs' fatigue, adding to the impossibility of their being able to relax and recharge. Therefore, it is possible that this complex period for HCPs influenced their experiences, especially when they mentioned the work overloads and fatigue that hindered the implementation of the interventions. The participants in our study consisted of a selection of HCPs who were working in a general medical rehabilitation unit, who agreed to be trained in TM, and who agreed to be interviewed in a focus group format. Thus, a selection bias may have occurred, and our results may not be applicable to all HCPs working in such an environment or in different settings. Hence, the results may be limited in terms of transferability [39]. As in all qualitative studies, our study sample was small, and the exploration of experiences in this specific population indicates that the transferability may thus be limited to people and settings with characteristics similar to those investigated in this study. Another challenge encountered in the study was focusing the care providers on their experiences with TM and not their global work assignments. This points to the difficulty of comprehending the satisfaction and hurdles related to only one specific task among a variety of tasks and responsibilities.

Finally, we cannot exclude that the appreciation of TM expressed by the participants was linked to the personal characteristics of the specialized nurse providing the TM training. Other experiences may thus be obtained with different styles of therapists.

Despite its limitations, the strength of this research is the size of the HCP group who participated in the study compared to the number of HCPs in each unit. This number further includes nurses and nursing auxiliaries, which led to a representative sample for each unit. Despite the various hurdles related not only to the pandemic but also to the perceived and actual conceptual and organizational barriers, the HCPs of the two units involved in this study were ready to participate and provide informed points of view regarding the object of this study.

## 5. Conclusion

Despite the perceived benefits of TM reported by the HCPs, ambivalence arose around the legitimacy of this intervention. The results of this study emphasize the importance of changing HCPs' attitudes regarding a given intervention in order to facilitate its implementation. Indeed, the perceived benefits do not seem to be enough to justify the use of an intervention, especially if the intervention is not perceived as being part of nursing care. Similar difficulties have been observed with CAM, despite HCPs' positive attitudes and its perceived benefits [16]. Consequently, the development and implementation of new interventions will be optimal only if HCPs feel legitimate in using them. According to our results, factors that could facilitate such process are increased availability of empirical evidence, increased knowledge of HCPs and supervisors through training, and inclusion of HCPs in the intervention's development process. Further, conceptual tensions and organizational barriers could be improved by integrating a global approach from the very beginning of HCPs' education.

## Acknowledgments

The authors want to thank all health care professionals who participated and shared their experiences in this study. We would also like to thank the research assistant Camille Thentz for her contribution to the data collection.

## Author Contributions

**Conceptualization:** Gora Da Rocha Rodrigues, Monique Boegli, Catherine Bollondi Pauly, Christophe Luthy, Jules Desmeules, Christine Cedraschi.

**Formal analysis:** Gora Da Rocha Rodrigues, Adrien Anex, François Curtin, Christine Cedraschi.

**Funding acquisition:** Gora Da Rocha Rodrigues, Monique Boegli, Catherine Bollondi Pauly, Jules Desmeules, Christine Cedraschi.

**Investigation:** Gora Da Rocha Rodrigues, Catherine Bollondi Pauly, François Curtin, Jules Desmeules, Christine Cedraschi.

**Methodology:** Gora Da Rocha Rodrigues, Monique Boegli, François Curtin, Jules Desmeules, Christine Cedraschi.

**Project administration:** Gora Da Rocha Rodrigues, Catherine Bollondi Pauly, Christophe Luthy, Jules Desmeules, Christine Cedraschi.

**Resources:** Gora Da Rocha Rodrigues.

**Software:** Gora Da Rocha Rodrigues.

**Supervision:** Gora Da Rocha Rodrigues, Monique Boegli.

**Validation:** Gora Da Rocha Rodrigues, Christophe Luthy, Jules Desmeules, Christine Cedraschi.

**Writing – original draft:** Gora Da Rocha Rodrigues, Adrien Anex, Christine Cedraschi.

**Writing – review & editing:** Monique Boegli, Catherine Bollondi Pauly, François Curtin, Christophe Luthy, Jules Desmeules.

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
