## [Decision Letter · Decision Letter 0]

13 Sep 2022

PONE-D-22-18019Is massage a legitimate part of nursing care? A qualitative studyPLOS ONE

Dear Dr. da Rocha Rodrigues,

Thank you for submitting your manuscript to PLOS ONE. After careful consideration, we feel that it has merit but does not fully meet PLOS ONE’s publication criteria as it currently stands. Therefore, we invite you to submit a revised version of the manuscript that addresses the points raised during the review process.

We look forward to receiving your revised manuscript.

Kind regards,

Maryam Farooqui, Ph.D

Academic Editor

PLOS ONE

Journal Requirements:

Reviewers' comments:

Reviewer's Responses to Questions

**Comments to the Author**

1. Is the manuscript technically sound, and do the data support the conclusions?

Reviewer #1: Yes

2. Has the statistical analysis been performed appropriately and rigorously? 

Reviewer #1: Yes

3. Have the authors made all data underlying the findings in their manuscript fully available?

Reviewer #1: No

4. Is the manuscript presented in an intelligible fashion and written in standard English?

Reviewer #1: Yes

5. Review Comments to the Author

Reviewer #1: Is massage a legitimate part of nursing care? A qualitative study

Comments to the Author

Congratulation on the submitted manuscript. The topic is timely and will be of interest to the readers of the journal. However, few changes are suggested to improve the clarity of this manuscript.

I have several recommendations and questions about the manuscript.

Comment

1. A qualitative study approach was used for this study involving how many participants? It is necessary to provide the total number of participants in your approach at the beginning of your statement.

-The inclusion criteria for the study need to be discussed here.

-List the Five themes that emerged from the thematic analysis in the abstract.

2. Need to explain the full meaning of The ICD-11 for the first use in a sentence.

3. Please harmonize the statements and explain more regarding this.

-An intervention (TM or control) was assigned to each of the two units/clusters.(page 6,para 134)

4. Explain the full meaning what is a unit (CG or IG).example CG=Control Group for the first of the statement.

5. Ethical consideration- Please add the reference number or approval code.

6. I would like to suggest to the author provide more information on the methodology of the study such as:

-How about

the sample size selected?

-The total number of respondents?

-The inclusion and exclusion criteria?

-The interview protocol process consisted of what?

To briefly explain the interview protocol for FGD. What were the questions asked in this phase?

7. All the participant’s statements must be italic

8. No figure attached

9. References: 10/39 of the references are more than 10 years. Recommendation: 5 years above is better.

6. PLOS authors have the option to publish the peer review history of their article (what does this mean?). If published, this will include your full peer review and any attached files.

Reviewer #1: No

---

## [Author Response · Author response to Decision Letter 0]

27 Oct 2022

Dear Editor and Reviewer,

We would like to thank you for the opportunity to revise our manuscript entitled “Is massage a legitimate part of nursing care? A qualitative study”. We appreciate the careful reviews and constructive suggestions we received. 

The manuscript has been revised to address the editor’s and reviewers’ comments and we believe the revisions have helped us to improve our manuscript. Original comments are listed below followed by our responses in italics. You will find enclosed a marked-up copy and unmarked version of the revised manuscript we would like to submit for publication in PLOS ONE as an original research. Each author has given approval to the final form of the revised manuscript. 

We hope that this revised manuscript can be accepted for resubmission in PLOS ONE. We remain receptive to any further suggestions that would improve the paper. Thank you for your consideration. We look forward to hearing from you.

Yours sincerely,

Reviewer #1: 

Congratulation on the submitted manuscript. The topic is timely and will be of interest to the readers of the journal. However, few changes are suggested to improve the clarity of this manuscript. I have several recommendations and questions about the manuscript.

Comment

Abstract

1. A qualitative study approach was used for this study involving how many participants? It is necessary to provide the total number of participants in your approach at the beginning of your statement.

 We have now clearly stated the numbers of participants in the focus groups that are at the core of this qualitative study (p. 2, para 48)

“10 caregivers from the TM group and 6 from the machine group”

-The inclusion criteria for the study need to be discussed here.

 HCPs were included in this study if they participated in the training about TM or the use of the machine delivering massage. Thus, we added it in the text accordingly (p. 2, para. 46). 

“At the end of the trial, two focus groups were conducted with HCPs from each unit who took part in the training and agreed to discuss their experience: 10 caregivers from the TM group and 6 from the machine group.”

-List the Five themes that emerged from the thematic analysis in the abstract.

This has been amended in the text accordingly (p. 2, para. 52)

“Five themes emerged from thematic content analysis: perceived impact on patients, HCPs’ affective and cognitive experiences, patient-professionals relationships, organizational tensions, and conceptual tensions.”

Introduction

2. Need to explain the full meaning of The ICD-11 for the first use in a sentence.

Indeed, it is important to specify it. Therefore, we added the full name this abbreviation (p. 4, para. 69)

“The International Classification of Diseases 11th Revision (ICD-11)”

Methodology

3. An intervention (TM or control) was assigned to each of the two units/clusters (page 6,para 134). Please harmonize the statements and explain more regarding this.

 We thank the reviewer for his/her comment which allows us to improve the overall clarity of our setting. Two units of a general rehabilitation ward from a university hospital in Switzerland were selected for this study. We compared TM with a machine-delivered massage, using a non-randomized cluster clinical trial design. Each of these two interventions were assigned to one of the units. For better clarity, we modified the text accordingly (p. 6, para. 132).

“The study was designed as a non-randomized cluster clinical trial with an exploratory qualitative part [32]. Two units of a general rehabilitation ward were selected for this trial. TM has been assigned to one unit (intervention group; IG) whereas machine-delivered massage has been assigned to the other (control group; CG).”

4. “The HCPs received training on TM or the use of the machine according to their unit (CG or IG)” (page 6, para 137). Explain the full meaning what is a unit (CG or IG). Example CG=Control Group for the first of the statement.

Please see previous comment (3) for a better description of what unit (CG or IG) means.

5. Ethical consideration- Please add the reference number or approval code.

This has been amended in the text as advised (p. 7, para. 139).

“This study received the approval of the Cantonal Commission for Ethics and Human Research in Geneva (CCER 2019-00848) and was pre-registered (ClinicalTrials.gov, NCT04295603) [32].”

6. I would like to suggest to the author provide more information on the methodology of the study such as:

-How about the sample size selected?

We have now tried to be more explicit regarding the selection of the participants to the 2 focus groups and the total number of respondents. We added in the text as advised (p. 7, para. 150).

“From the 21 HCPs working in the IG unit, 15 HCPs (7 nurses and 8 nursing auxiliaries) took part in the TM training and in the trial. In the CG unit all 21 HCPs working in this unit were offered participation, and 8 of them took part in the machine-delivered massage training and in the trial”

-The inclusion and exclusion criteria?

 Inclusion criteria were: to be working in one of the two units selected for this study and to have taken part to the training on either TM or the use of the machine. We modified the text accordingly (p. 7, para. 148)

“HCPs were included if they were working in one of the two selected units and participated in the trainings whether or not they delivered the interventions. We aimed to include as many opinions as possible as our goal was to investigate the feasibility of the interventions.”

-The interview protocol process consisted of what?

We have now added a paragraph regarding the choice of our investigation method (p. 7, para. 159)

“Focus groups have been chosen to explore the satisfaction and general perception of massage in the multidisciplinary health care teams in the two units concerned. The choice of this method was of relevance in the context of the various units of the ward that all function as specific teams. Thus, a method of data collection that simultaneously generates data for three levels of analysis: the individual, the group and the interactions between participants was of clear interest. The protocol insisted on the need to recruit at least five participants, including nurses, assistant nurses, physical therapists and/or physicians. Based on the experience of our research team regarding TM, an interview guide was devised to assess the impact of massage on general care, the experience thereof, the positive and negative effects, and its impact on the development and planning of care (see S1 Table). The interview has been audio-recorded, and transcribed verbatim.”

- To briefly explain the interview protocol for focus group discussion (FGD). What were the questions asked in this phase?

We thank the reviewer for his/her suggestions. Therefore, we added a brief description of the dimensions explored in the interview guide and give examples of questions (p. 8, para. 170) 

“The interview guide explored the HCPs’ experiences and use of the interventions (see Table 1). Four dimensions were investigated : recall of the massage, general appreciation of massage, facilitators and barriers in the experience of massage, and benefits for other.” 

7. All the participant’s statements must be italic

This has been amended in the manuscript as advised.

8. “Those themes were regrouped into two dimensions: outcomes of the interventions and implementation in nursing care (see Figure)”. No figure attached

Indeed, following the submission guidelines of Plos One, Figures should not be included in the manuscript but should be uploaded as a separate document. Thus, we included only the figure caption in the manuscript. For your information, figure 1 is attached below:

9. References: 10/39 of the references are more than 10 years. Recommendation: 5 years above is better.

 We agree with the reviewer. However, some of the articles are at the foundation of a theory (e.g. Braun and Clarke, 2006 regarding thematic analysis; or Rogers, 1970, and Newman, 2008, regarding the theoretical basis of nursing), or are regarded as landmarks in their field (e.g. Malterud, 2001 challenges and guidelines for the use of qualitative methods outside of social sciences; or Kitzinger, 1994 which still provides a very well-informed methodology for focus groups) or we just could not find more recent article. We carefully checked that our “old” articles could not be replaced by more recent ones.

---

## [Decision Letter · Decision Letter 1]

16 Jan 2023

Is massage a legitimate part of nursing care? A qualitative study

PONE-D-22-18019R1

Dear Dr. da Rocha Rodrigues,

We’re pleased to inform you that your manuscript has been judged scientifically suitable for publication and will be formally accepted for publication once it meets all outstanding technical requirements.

Kind regards,

Nabeel Al-Yateem, PhD

Academic Editor

PLOS ONE

Additional Editor Comments (optional):

Reviewers' comments:

Reviewer's Responses to Questions

**Comments to the Author**

1. If the authors have adequately addressed your comments raised in a previous round of review and you feel that this manuscript is now acceptable for publication, you may indicate that here to bypass the “Comments to the Author” section, enter your conflict of interest statement in the “Confidential to Editor” section, and submit your "Accept" recommendation.

Reviewer #1: All comments have been addressed

Reviewer #2: All comments have been addressed

2. Is the manuscript technically sound, and do the data support the conclusions?

Reviewer #1: Yes

Reviewer #2: Yes

3. Has the statistical analysis been performed appropriately and rigorously? 

Reviewer #1: Yes

Reviewer #2: N/A

4. Have the authors made all data underlying the findings in their manuscript fully available?

Reviewer #1: Yes

Reviewer #2: Yes

5. Is the manuscript presented in an intelligible fashion and written in standard English?

Reviewer #1: Yes

Reviewer #2: Yes

6. Review Comments to the Author

Reviewer #1: I acknowledge and have previously reviewed the corrections made by the author in the manuscript titled "Is massage a legitimate part of nursing care? A qualitative study" in accordance with the reviewer's suggestion. The reviewer’s recommendation is ACCEPT.

Reviewer #2: The authors have done an excellent job at responding to the reviewers questions and comments. I wish to thank them for addressing all the concerns and questions of the previous reviewers

7. PLOS authors have the option to publish the peer review history of their article (what does this mean?). If published, this will include your full peer review and any attached files.

Reviewer #1: **Yes: **RUSNANI AB LATIF(PhD in nursing)

Reviewer #2: No

---

## [Editor Report · Acceptance letter]

20 Jan 2023

PONE-D-22-18019R1 

Is massage a legitimate part of nursing care? A qualitative study 

Dear Dr. Da Rocha Rodrigues:

I'm pleased to inform you that your manuscript has been deemed suitable for publication in PLOS ONE. Congratulations! Your manuscript is now with our production department. 

Kind regards, 

on behalf of

Dr. Nabeel Al-Yateem 

Academic Editor

PLOS ONE